# Understanding the dynamics of obesity prevention policy decision-making using a systems perspective: A case study of Healthy Together Victoria

**Brydie Clarke**[1], **Janelle Kwon**[1], **Boyd Swinburn**[2], **Gary Sacks**[1]*

**1** Global Obesity Centre, School of Health and Social Development, Institute for Health Transformation, Deakin University, Geelong, Australia, **2** School of Population Health, University of Auckland, Auckland, New Zealand

* gary.sacks@deakin.edu.au

## Abstract

### Introduction

Despite global recommendations for governments to implement a comprehensive suite of policies to address obesity, policy adoption has been deficient globally. This paper utilised political science theory and systems thinking methods to examine the dynamics underlying decisions regarding obesity prevention policy adoption within the context of the Australian state government initiative, Healthy Together Victoria (HTV) (2011–2016). The aim was to understand key influences on policy processes, and to identify potential opportunities to increase the adoption of recommended policies.

### Methods

Data describing government processes in relation to the adoption of six policy interventions considered as part of HTV were collected using interviews (n = 57), document analyses (n = 568) and field note observations. The data were analysed using multiple political science theories. A systematic method was then used to develop a Causal Loop Diagram (CLD) for each policy intervention. A simplified meta-CLD was generated from synthesis of common elements across each of the six policy interventions.

### Results

The dynamics of policy change could be explained using a series of feedback loops. Five interconnected balancing loops served to reduce the propensity for policy change. These pertained to an organisational norm of risk aversion, and the complexity resulting from a whole-of-government policy approach and in-depth stakeholder consultation. However, seven virtuous reinforcing loops helped overcome policy resistance through policy actor capabilities that were improved over time as policy actors gained experience in advocating for change.

**Data Availability Statement:** All relevant data are within the manuscript and its Supporting Information files.

**Funding:** GS was supported by a Heart Foundation Future Leader Fellowship (102035) from the National Heart Foundation of Australia (https://www.heartfoundation.org.au/). GS and BS are researchers within a National Health and Medical Research Council (NHMRC) (https://www.nhmrc.gov.au/) Centre of Research Excellence in Food Retail Environments for Health (RE-FRESH) (APP1152968) (Australia). GS and BS are also researchers within a NHMRC Centre for Research Excellence entitled Reducing Salt Intake Using Food Policy Interventions (APP1117300). The authors are solely responsible for the opinions, hypotheses and conclusions or recommendations expressed in this publication, and they do not necessarily reflect their funders' vision. The funders had no role in study design, data collection and analysis, decision to publish, or preparation of the manuscript.

**Competing interests:** The authors have declared that no competing interests exist.

## Conclusion

Policy processes for obesity prevention are complex and resistant to change. In order to increase adoption of recommended policies, several capabilities of policy actors, including policy skills, political astuteness, negotiation skills and consensus building, should be fostered and strengthened. Strategies to facilitate effective and broad-based consultation, both across and external to government, need to be implemented in ways that do not result in substantial delays in the policy process.

## Introduction

The prevalence of overweight and obesity continues to increase internationally with significant concomitant impacts on morbidity and mortality [1]. The causes of obesity are complex, with numerous individual, social and environmental factors identified as playing a role in the etiology of overweight and obesity [2,3]. While there is some debate among stakeholder groups as to the most appropriate response to rising rates of obesity [4,5], it is generally agreed that comprehensive, multi-sectoral approaches, across multiple levels of government, are required [6,7]. There is also strong evidence that many policy options are likely to be effective and cost-effective ways to support obesity prevention efforts at the population-level [8]. Whilst some countries have recently increased the use and widened the scope of policy interventions to reduce obesity [9,10], progress has been "patchy" [8]. To date, implemented policies have largely been directed towards influencing individual physical activity- and diet-related knowledge and behaviors through 'soft' policy options, such as nutrition education programs and social marketing campaigns, whilst regulatory responses have been utilised to a lesser extent [11,12].

In order to advance the breadth and scope of obesity prevention efforts, it is important to understand the barriers and enablers of obesity prevention policy action. Previous studies have identified several underlying challenges for successful adoption of recommended policies, including limited skills, knowledge and capabilities of policy actors [13,14], food industry resistance and lobbying [15–17], and socio-political factors that shape policy maker preferences [18,19]. Whilst previous studies identify potential determinants of policy adoption, they do not provide a comprehensive understanding of the complexity of policy processes, nor the dynamics of how each of these elements relate to one another [20].

Political science theorists identify the need to consider policy decision-making through non-linear perspectives [21–23], appreciating the multiple, interacting forces guiding policy decisions [24]. Additionally, some authors have contended that utilisation of a systems thinking perspective can add further value to theoretical accounts of policy processes by specifying the relationship between mechanisms of change [24,25]. A systems thinking approach requires a holistic perspective, bringing together consideration of underlying structures and patterns and how these influence the behavior of a system as a whole [26,27]. A systems thinking approach also acknowledges that system behavior is usually governed by feedback processes which can be either *reinforcing* (for example, money in a savings account will generate interest, which increases the balance in the savings account and earns more interest) [28] or *balancing* (for example, the human body sends a signal, through hormonal and nerve signals, from the stomach to the brain when food is eaten to appease the feeling of hunger) [28]. Reinforcing loops may be virtuous or vicious in terms of their impact on system behaviour [29]. Virtuous

loops involve the magnification of positive change (e.g., increase in staff skills) and the reduction of negative factors (e.g., staff turnover). Whereas, vicious cycles involve the amplification of detrimental changes and the decline of positive changes [29]. In contrast, balancing feedback loops work to stabilise systems, by limiting growth and slowing decay. Causal Loop Diagrams (CLDs) are one systems thinking method to help document the feedback mechanisms and interconnections between system components [30]. CLDs can be a useful heuristic tool to highlight virtuous or vicious reinforcing feedback dynamics [31], and identify leverage points by which strategies came to be implemented to alter system behaviour [32,33].

While systems thinking and CLDs have been utilised extensively to understand the causes and complex interdependencies driving obesity prevalence [34,35], and to inform and evaluate obesity prevention initiatives [29,36], the application to policymaking within the obesity prevention context is scarce [37]. This paper utilised political science theory and CLD methods to examine the dynamics underlying decisions regarding obesity prevention policy adoption within the context of the Australian state government initiative, Healthy Together Victoria (HTV). The aim was to understand key influences on policy processes, and to identify potential opportunities to increase the adoption of recommended obesity prevention policies.

## Methods

### Study setting

Healthy Together Victoria (HTV) was a major state government-led initiative implemented in Victoria, Australia between 2011–2016, which aimed to deliver a multi-level, multi-setting complex systems approach to obesity prevention [38,39]. HTV represented a significant investment in obesity prevention by the Victorian Department of Health and Human Services (DHHS), with the vast majority of funding for HTV coming from the Australian Commonwealth Government through the now defunct National Partnership Agreement on Preventative Health [40]. HTV focused on environmental and underlying structural changes to support obesity-related behaviour change at a population level. As such, there was a considerable policy effort as part of the initiative, including a range of policy instruments implemented at both the state and local government level. At the local level, HTV funding was used to employ a large workforce of health promotion practitioners with the goal of leveraging opportunities for obesity prevention across the community. Health promotion practitioners funded under HTV were charged with mobilising local action and policy change. This involved various initiatives across numerous settings including schools, workplaces and healthcare organisations. The initiative consisted of 12 implementation sites (Healthy Together Communities (HTCs)) across 14 local government areas. HTC sites were selected on the basis of high need, with a focus on areas with elevated levels of overweight, obesity and other factors contributing to chronic disease risk. The relatively large scale and diversity of the policy change effort that occurred during the implementation of HTV provided an important opportunity to understand the system dynamics of policy change with respect to obesity prevention.

### Study design

The research used a single embedded case study design based on the methods described by Yin [41]. Six distinct interventions that policy makers considered for adoption as part of HTV served as embedded units within the case study. The policy interventions selected for inclusion were derived from the broader suite of policies considered for adoption as part of HTV, with selection based on key characteristics of each intervention to ensure heterogeneity (e.g., in terms of policy instrument type) and to provide the greatest opportunity to learn [41,42]. Details of the six selected policy processes investigated are provided in Table 1. Two of the

**Table 1. Summary of Healthy Together Victoria policy processes investigated.**

| Policy | Description | Policy development period analysed | Date of policy adoption |
|---|---|---|---|
| *LiveLighter®* | *LiveLighter®* is a healthy lifestyle promotion and education program developed in Western Australia. The content of the *LiveLighter®* program featured, amongst other aspects, 'toxic fat' and other graphic imagery designed to shock individuals and promote healthier eating behaviours. | 2010–2014 | April 2014 |
| *The Achievement Program* | A quality framework, associated benchmarks and funding for workforce support to enable settings (e.g., schools and workplaces) to create healthier environments across a number of priority areas (including healthy eating and physical activity, amongst others). | 2008–2012 | August 2011 |
| *Jamie's Ministry of Food* | A community cooking program that aimed to improve program participants' skills, knowledge and confidence in cooking and to encourage healthy eating. | 2009–2012 | January 2012 |
| *Menu Kilojoule Labelling Legislation* | Legislation that required chain fast food outlets to display energy content (in kJ) of standard, ready-to-eat food and non-alcoholic drinks on menu boards to support consumers to make informed healthier choices. | 2009–2017 | February 2017 |
| *Healthy Catering Policies* | Catering or procurement policies with a healthy food focus, adopted by the Victorian state government and four local governments. | 2009–2017 (state) | October 2016 (state) |
| | | 2013–2015 (LG1) | September 2015 (LG1) |
| | | 2012–2015 (LG2) | September 2015 (LG2) |
| | | 2012–2013 (LG3) | July 2013 (LG3) |
| | | 2013–2014 (LG4) | July 2014 (LG4) |
| *Land Use Planning Policies* | Victorian state and local government land use policies that supported obesity prevention efforts, with a focus on physical activity, active recreation and restrictions on fast food outlets. | 2009–2015 (state) | March 2017 (state) |
| | | 2013–2016 (LG1) | November 2016 (LG1) |
| | | 2013–2016 (LG2) | March 2016 (LG2) |
| | | 2013–2017 (LG3) | May 2017 (LG3) |
| | | 2014–2017 (LG4) | May 2017 (LG4) |

Notes: LG 1, LG 2 etc. represents the four local government area policies investigated as part of the analyses, with each local area assigned a sequential identifying code.

policies (*Healthy Catering Policies* and *Land Use Planning Policies*) involved policy processes at both local and state government levels. The local-level analyses were conducted within four of the 12 HTC sites that were in place during the implementation of these interventions. The selection of HTC sites was based on consultations with HTV policy makers, who indicated that these sites would provide the most variety of policy processes and experiences [42,43]. All of the selected policy interventions were adopted in Victoria over the period 2011–2017, although some policies, such as *Menu Kilojoule Labelling Legislation* required multiple formal iterations of policy proposals prior to policy adoption, and some aspects of the *Land Use Planning Policies* (e.g., those pertaining to fast food outlets) were not adopted.

## Data collection

Qualitative data from each embedded unit within the case study included in-depth semi-structured interviews (n = 57) with participants involved in the policy development and decision-making processes, documents (including public records of parliamentary debates, available policy reports and media articles, n = 568), and field note observations. All data collection was conducted by one of the researchers (BC) whilst on a research placement in the Obesity Prevention Unit of the DHHS, Victoria, between 2015–2018. This enabled the research to be developed and conducted within the 'real-world' policy context [44].

**Interviews.**  Key informant interviews were conducted between December 2015 and April 2017. Participants were identified through purposive sampling, supplemented by snowball sampling [45]. Purposive sampling was facilitated by the positioning of one of the researchers (BC) in the policy context, which helped with identification of relevant and appropriate interview participants through the researcher's direct observations and interactions with policy makers. Participants were selected based on their anticipated ability to provide detailed insight and first-hand experience into the policy processes related to each of the included HTV policy interventions (assessed separately for each policy). The total number of interviewees conducted in relation to each of the HTV policy interventions ranged from six (*Jamie's Ministry of Food*) to 24 (*Land Use Planning Policies*) as shown in **S1 Appendix**

The varying number of interviewees for each policy reflects the differing scale, prominence and number of stakeholders involved in the policy processes regarding each of the HTV initiatives. Participants included: current and former politicians (n = 4); senior ministerial staff (n = 2); government officials at state (n = 26) and local levels (n = 12); academics (n = 4); and senior representatives from key public health organisations (n = 7) and private sector organisations (n = 2), such as food retailers. All but one of the potential interviewees who were invited to participate accepted in relation to each of *The Achievement Program*, *Jamie's Ministry of Food*, *LiveLighter*® and *Healthy Catering Policies* investigations, whilst two identified interviewees did not participate in relation to *Menu Kilojoule Labelling Legislation* and *Land Use Planning Policy* (n = 6 in total). There were also a small number of potential interviewees (n = 4), identified through the snowball sampling process, who were not contactable in relation to *The Achievement Program*, *Jamie's Ministry of Food* and *Menu Kilojoule Labelling Legislation*, as they had either left their respective organisations or were on extended leave at the time of data collection.

Interviews were guided by a semi-structured interview schedule (**S2 Appendix**), which was informed by a review of the literature and the political science frameworks underpinning the study [46]. Participants selected the location of the interviews, which occurred either within a workplace or neutral public setting (e.g., café), except for four participants who elected to participate by phone interview. The duration of interviews ranged from 16 minutes to 75 minutes. Whilst most participants were asked about one specific HTV policy intervention, some interviewees provided insights in relation to multiple policies. Eight participants from the local government level were interviewed in relation to both the *Healthy Catering Policies* and *Land Use Planning Policies*.

**Documents.**  Documents relevant to the decision-making processes of the policies under investigation were obtained through DHHS contacts. Document collection was facilitated by the embedded nature of the research and occurred in an iterative manner, with interviewees asked whether they could suggest any documentation that may be relevant to the study.

Documents included internal policy briefings, reports, consultation papers, evaluation reports and other documents. Additional searches were undertaken to identify relevant public documents of the Victorian Hansard database, which contains public records of parliamentary debates, and of the Factiva database for public media reports and documents. Further details of the documents analysed are provided in **S3 Appendix**

As with the interviews, the varying number of documents analysed for each policy reflects the differing characteristics and scale of the policy processes regarding each of the HTV initiatives.

**Observations.**  Open ended narrative field notes were captured by one researcher (BC) through observing and taking notes on daily work-related activities within the HTV policy context from April 2015 until December 2017 [47]. This 'overt' observation [45] allowed examination of how public health practitioners and policy actors worked to influence policy related

to obesity prevention. All members of the Obesity Prevention Unit consented to this observation. Organisational consent was also obtained.

## Data analysis

A two-stage analysis process was undertaken. In the first stage, data were analysed for each of the selected policy interventions separately. As part of this first stage of analysis, a CLD was generated to describe the dynamics of the policy process for each intervention. In the second stage, the six individual CLDs were synthesised to produce a meta-CLD of the HTV policy system based on themes that were consistent across the six CLDs. All data analysis was initially performed by one of the researchers (BC), and reviewed by another (GS), with queries resolved in consultation with a third researcher (BS) where necessary. Further details of each stage of the analysis are provided below and a schematic overview of the study analysis approach is displayed in **Fig 1**.

**Analysis of individual policy processes.** The analysis of the policy process for each policy intervention involved a deductive thematic analytical approach underpinned by multiple political science theories. The theories employed were the Advocacy Coalition Framework (ACF) supported by the Institutional and Analysis Framework (IADF), and the Multiple Streams Theory (MST). These theories were chosen based on a systematic review of the application of political science theory to the study of obesity prevention policy processes, which identified the value of each theory and recommended employing multiple theoretical perspectives in obesity prevention policy analysis [46]. Data (interviews, documents and field notes) were systematically examined and coded against key constructs of the selected theories using a defined codebook (available on request), with coding facilitated by the qualitative software NVivo10. Then, as described by Kim and Andersen [32], coded data were 'micro-analysed' to identify cause and effect variables, and the relationships between variables. This process involved iteratively documenting variable names, direction of linkages between elements and the association direction (i.e., polarity). These causal structures were used to develop the final graphical representation of the CLDs using Vensim software (Ventana Systems, Harvard, MA). Detailed findings of the individual analyses are reported elsewhere [48–51].

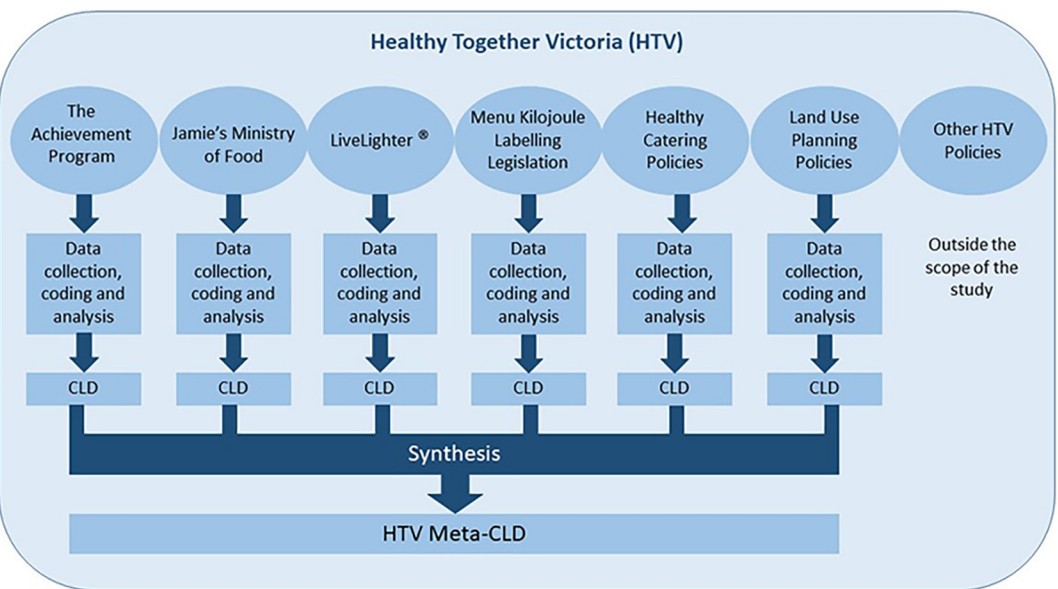

CLD = Causal Loop Diagram

**Fig 1. Overview of analysis process of the Healthy Together Victoria case study.**

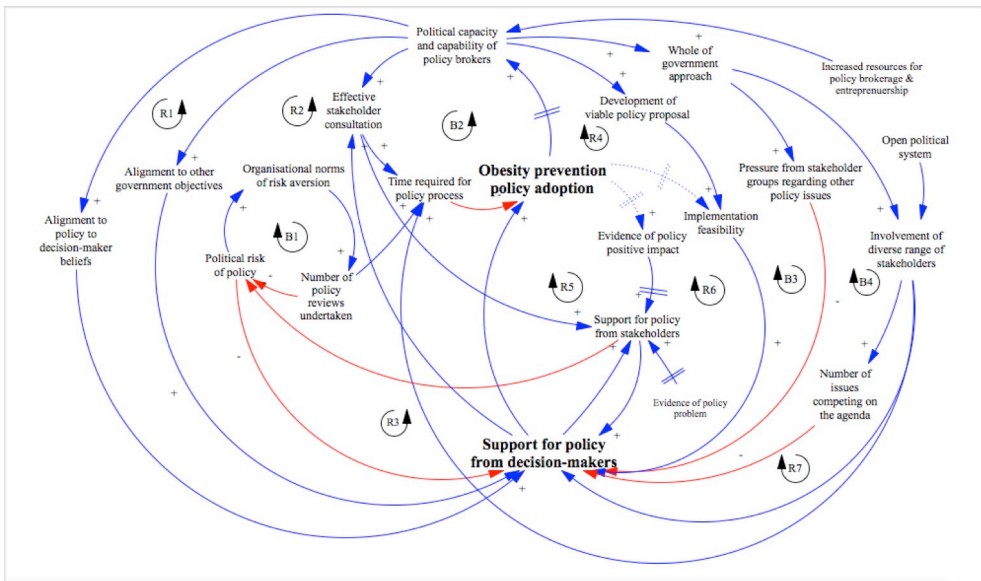

**Fig 2. Causal loop diagram of the Healthy Together Victoria policy process.** R loops (e.g., R1, R2 etc.) represent 'reinforcing loops' that magnify the effect of actors within the loop, including either positive or negative effects. B loops (e.g., B1, B2 etc.) represent 'balancing loops', in which feedback acts to stabilise the effect on the system. R loops and B loops were assigned sequential numbering as they were identified. A positive polarity (represented as '+') indicates that as a cause increases, the effect increases, and as cause decreases, the effect decreases. A negative polarity ('-') indicates an inverse relationship between the two variables (i.e., as cause increases, the effect decreases). A dash sign (//) indicates some delay in effect, relative to the time scale of the remainder of the diagram. Solid lines indicate that data indicated an association between factors, with triangulation across both data sources and methods. Dashed lines indicate where there was some data to demonstrate an association between factors, however triangulation across data sources or methods could not be achieved. **R1** = Policy actor capability to align policy interventions to decision-maker beliefs; **R2** = Policy actor capability to align policy interventions to other government objectives; **R3** = Stakeholder consultation as part of policy development; **R4** = Policy viability; **R5** = Evidence of policy impact; **R6** = Evidence of implementation feasibility; **R7** = 'Softening up' key stakeholders. **B1** = Organisational culture of risk aversion; **B2** = Time need for effective consultation; **B3** = Pressure from competing policy issues; **B4** = Involvement of a diverse range of stakeholders; **B5** = Time required to conduct a whole-of-government approach.

**Analysis to produce a meta-CLD.** To enable understanding of common influences on policy processes and opportunities for change within obesity prevention policy processes, a process of comparing the six individual HTV policy system CLDs was undertaken to identify recurrent associations and feedback mechanisms between system components. A meta-CLD of the HTV policy system was produced that represented system dynamics that were consistent across each of the six individual CLDs. The causal relationships and system feedback structures were documented in a simplified manner to enable readability and in an effort to communicate insights effectively [52]. The synthesis process was undertaken using an iterative approach to ensure the CLD was consistent with the findings from the qualitative data [53].

## Ethics approval

Ethical approval for the study was granted by the Deakin University Human Research Ethics Committee 2015 (HEAG-H 106_2015). Approval for the conduct of the research was also granted by the DHHS in 2015.

## Results

The analysis of the processes leading to the adoption of each policy intervention revealed numerous interconnected influences on obesity prevention policy processes [48,49,51].

Synthesis of the findings from each policy intervention revealed multiple recurrent associations and feedback mechanisms between components of the policy systems. These relationships are outlined in the meta-CLD of the HTV policy systems (**Fig 2**) using standard system dynamics notation [54]. The meta-CLD contained 12 influential feedback loops that provide insight into the complex dynamics of policy decision-making in the context of HTV. Five interconnected balancing loops served to reduce the propensity for policy change. However, seven reinforcing feedback loops strengthened over time to help drive policy change in relation to several policy interventions. Each of the loops are further explained below.

### Detail of feedback loops

**Reinforcing loops.** *R1 and R2*: *Increasing policy actor capacity and capability*. Synthesis of the thematic analyses, underpinned by the MST and ACF theories, helped to identify the importance of policy actors (either inside or outside of government) who played entrepreneurial [22,55] or brokerage [56,57] roles to secure policy adoption. Government policy makers and external policy actors that were able to frame policy narratives to *align policy proposals to decision-maker beliefs* (R1) played an important role in reinforcing policy change. According to interview participants, these actors commonly framed policies as part of a broader suite of complementary initiatives under HTV, particularly where there was criticism of the likely impact of a single policy. Policy documentation and interviews demonstrated how, in some instances, such as the *Menu Kilojoule Labelling Legislation*, policy makers also emphasised the ways in which the policy supported informed consumer choice, so as to align with the predominant political ideologies of neoliberalism and individual responsibility that were current at the time [48]. This strategic alignment of policies to decision-maker beliefs acted to increase support for policy intervention from decision-makers, in turn, leading to increased adoption of obesity prevention policies. Critically, the level of capability and use of these strategic skills by policy actors increased with each successful policy adoption over time, thus creating a virtuous reinforcing loop. For example, in interviews, policy makers reflected on how, during the development of *Menu Kilojoule Labelling Legislation*, they learnt to adapt policy framing to align with decision-maker beliefs and increase decision-maker acceptability.

> "... *The framing around the policy did get a bit more focused in this third [iteration of the policy] around providing Victorians with informed choice. So that kind of wording was quite strong in this government's, in this Minister's speaking. So that was a kind of transformation of the framing*"

> (DHHS Senior Policy Advisor 1)

A related reinforcing loop was the *alignment of policy proposals to other government objectives* (R2) by policy actors. This alignment of policy proposals to other government objectives was an important driver of decision-maker support for several HTV policy interventions. For example, *Jamie's Ministry of Food* was framed in policy documentation as aligning with broader preventive health policies, such as the Victorian Public Health and Wellbeing Plan, and government objectives related to increased productivity and reduced healthcare costs.

> "There is no 'silver bullet' to reducing the prevalence of obesity. JMoF [Jamie's Ministry of Food] can work alongside other government and non-government health initiatives and add value."

> (DHHS policy documentation, 2011)

These positive associations became virtuous reinforcing loops when the effectiveness of these framing tactics was shared with other public health policy staff or when continuity of staff allowed relevant policy actors to implement learnt successful tactics in future policy processes.

*R3*: *Stakeholder consultation as part of policy development*. The case study revealed that government staff involved in crafting policy proposals sought involvement from a diverse range of stakeholders as part of policy development processes. Interview participants noted that decision-makers recognised the advantages of stakeholder consultation in policy development, particularly for the *LiveLighter*®, *Menu Kilojoule Labelling Legislation*, and the *Achievement Program* policy processes, whereby numerous stakeholder groups who would be impacted by the policy were reportedly made to feel that they had an opportunity to contribute to the policy development process [48,49,51]. In addition, stakeholder groups were given the opportunity to learn, at an early stage of the process, about how the proposed policy was likely to impact them [58,59]. Across the HTV policies investigated, consultations resulted in greater commitment to the success of the policy (or at least acquiescence), and, often, shared ownership of success [22]. Consequently, according to interview participants, stakeholder consultations reduced the perceived political risks associated with policy adoption, leading to increased support from decision-makers, and thus creating a reinforcing loop (R3). Indeed, a former Minister identified stakeholder consultation as a critical facilitator in the adoption of the *LiveLighter*® campaign.

> *"I think part of [the reason that the policy was adopted] was the process of consultation itself, actually. Often people then feel involved, and they have a deeper understanding of what's going on, what's trying to be achieved, any potential negatives and how you might manage those. I think [the facilitating factor] was doing [the consultations]"*

(Former Minister 1)

*R4*: *Policy viability*. The MST emphasises the importance of developing viable policy solutions in order for approval to be granted by decision-makers [22]. According to interviewees, policy adoption was enhanced when policy actors developed viable policy proposals that met the requirements of decision-makers. In the case of legislated policy (e.g., *Menu Kilojoule Labelling Legislation*), prescriptive documentation requirements were met [48]. For several other policy interventions, evidence from relevant contexts (e.g., tobacco policy or other Australian jurisdictions) was utilised to demonstrate implementation feasibility [48,51]. The presentation of viable and feasible policy solutions was described by participants as important in increasing support for policy interventions from decision-makers, leading to increased obesity prevention policy adoption. Over time, successful implementation of HTV policies increased the capacity and capability of policy actors to develop viable solutions, thus creating a virtuous loop (R4).

*R5 and R6*: *Utilisation of policy evidence*. The ACF's notion of 'policy-oriented learning', whereby new evidence can shape decision-maker beliefs and result in policy change, was helpful in highlighting the role that evidence played in policy adoption [57]. Interviews and documents revealed that evidence of various forms were utilised in the HTV policy processes. *Evidence of policy impact* (R5), particularly evidence generated through evaluations of 'real-world' obesity prevention policy adoption and implementation, was identified as an important part of securing decision-maker support [60]. According to participants, evidence from evaluations of implementation elsewhere in Australia was used to demonstrate the likely positive impact and increase acceptability of several HTV policies. For example, evidence of the impact

of the *LiveLighter®* campaign in Western Australia was reported to have helped convince decision-makers that the policy was likely to be effective in Victoria.

> *"We've seen the [Western Australia] material [for LiveLighter®], and we're able to adapt that. . . There was good research behind [the LiveLighter® policy] to suggest that it may well be effective"*

(Former Minister 1)

In addition, *evidence of implementation feasibility* (R6), often independent of evidence of policy effectiveness, was noted by participants as critical in regards to several policy interventions, including the *Achievement Program*, *Jamie's Ministry of Food*, *Live Lighter®*, and *Menu Kilojoule Labelling Legislation* [48,49,51]. An example of the use of evidence of implementation feasibility was in relation to *Menu Kilojoule Labelling Legislation*, where government policy makers introducing the policy emphasised the established nature of similar policies in other Australian jurisdictions.

> *"Kilojoule labelling laws have been in operating for nearly five years now in New South Wales. The scheme is well understood and accepted by industry. Similarly, we expect a smooth transition to the laws in Victoria. It is estimated that 80 per cent of the businesses required by the legislation to display kilojoule content area already doing so in other states. Therefore, the measurements have already been calculated and will be easily available"*

(Victorian Parliament Hansard, Legislative Council, 13 October 2016)

This use of evidence of policy impact was in addition to the provision of evidence of the policy problem as part of almost all documentation with respect to each policy intervention. For example, evidence of the high prevalence of obesity, and its health and economic consequences, was frequently cited in policy documentation. The various forms of evidence were important in increasing the support for obesity prevention policy options among external stakeholders and decision-makers. These virtuous reinforcing loops (R5 and R6) contributed to policy adoption in all the HTV policies, except for the *Land Use Planning Policies*, for which there was insufficient evidence regarding the causal association between fast food outlet density and obesity to convince decision-makers of the need for policy change in respect of fast food outlets.

*R7*: *'Softening-up' of key stakeholders*. The MST theory suggests that 'softening up' processes, whereby policy proposals are discussed with stakeholders who have to 'soften' up to new policy ideas, is an important method to increase the acceptability of policy solutions [22, p. 123]. There was strong evidence from interviews for this effect across the HTV policy process investigations, with examples of internal government staff members facilitating network coordination and negotiation processes with stakeholders to reduce resistance to policy options. In the adoption of the *Achievement Program*, policy makers commented on how they strategically engaged stakeholders from the Department of Education early and throughout the policy process in order to increase their support.

> *"[The] original steering group only had two representatives from the Department of Health and eight from Department of Education. We deliberately sought to shift some of that power to Education as a strategy to engage them, to give them a bit more power and control over the direction to inform it right from the beginning"*

(DHHS Senior Policy Advisor 2)

Interviewees identified that the support for a policy from key stakeholders helped to reduce the political risks of that policy, which, in turn, increased the support for the policy from decision-makers. The processes regarding the adoption of *LiveLighter®* also demonstrated how some decision-makers (e.g., a Minister) who supported the policy also worked to soften-up other stakeholders and decision-makers (e.g., other members of parliament) in order to increase support and reduce the potential for backlash [51]. This process created a virtuous reinforcing loop over time (R7) to increase support for obesity prevention policy adoption.

**Balancing loops.**   *B1*: *Organisational culture of risk aversion*. The ACF analysis, supported by the IADF, was particularly useful in highlighting the role of the organisational norm of risk aversion in resisting policy change. Across HTV policy process investigations, organisational risk aversion resulted in an increased number of policy proposal reviews, undertaken to reduce the perceived political risk of policy change. This risk aversion was particularly evident for the more controversial *LiveLighter®* social marketing campaign and the *Menu Kilojoule Labelling Legislation* whereby additional consultation or modified policy proposals were required to overcome perceived risks [48,51]. In relation to the *Land Use Planning Policies*, interview participants reflected on how organisational risk aversion increased reluctance to attempt policy changes unless they were aligned with decision-maker attitudes.

> *"We were going to do a policy around limiting fast food outlets and put in a Land Use Planning Policy review but we felt that [the local government area] wasn't really ready to listen to that. . . because what's happened in the past is that things have tried to get into the review of Land Use Planning Policy, even around food, and it's been 'chucked out'. It has got up to the council and a particular Councillor said "you can't restrict that sort of stuff" and took it out"*

> (Local Government Manager 1)

A consequence of the increased number of policy reviews was that these processes increased time for policy change. Interviewees indicated that this created a barrier to policy adoption due to the relatively short-term (4 year) political cycles in Victoria, especially considering that there were changes in the governing political party over the course of the policy study period. The reduced likelihood of policy change, over time, was related to both the shifts in political inclinations of leadership and the flow-on effects that the political party turnover had on the institutional structures.

*B2*: *Time needed for effective consultation*. As HTV progressed, experience in successful obesity prevention policy adoption helped increase the capability of policy makers in regard to effective stakeholder consultations. As noted in regards to R3, these capabilities worked to reduce political risks and garner policy support among decision-makers. However, according to interview participants, consultation processes elicited opposition to policy proposals from several different stakeholder groups, including the food industry (in relation to *Menu Kilojoule Labelling Legislation*), the construction industry (in relation to *Land Use Planning Policies*) and mental health groups (in relation to *LiveLighter®*) [48,51]. Interviewees indicated that stakeholder consultation processes resulted in substantially increased time required for policy change, which, as identified earlier (B1), presented a risk to policy adoption. For example, in the case of the *Land Use Planning Policies*, it was reported that the requirement for public exhibition and consultation increased the complexity of policy change and the likelihood of delayed policy decisions.

> *"The [strategic planning team] are still dealing with the [public] submissions they've had from the [land use planning policy] exhibition, and I think they're trying to avoid going to Panel [a*

*Planning Panel hearing–a formal process whereby a planning authority reviews proposed amendments]. The trouble with planning scheme amendments is that if one person makes a submission and they won't withdraw it–even if it's irrelevant, you can end up at a panel hearing"*

(Local Government Policy Officer 1)

The time needed for effective consultations thus served as a balancing loop that can help explain the policy resistance that needed to be overcome in order for policy change to occur.

*B3*, *B4 and B5*: *Consequences of a whole-of-government approach*. The theoretical analysis, and in particular the IADF, helped illuminate how the provision of increased resources for policy processes, acquired through the HTV funding, enhanced the capacity of obesity prevention policy advocates to engage in whole-of-government policy processes. This was particularly evident at the local government level, where there has historically been limited funding for health promotion workforce capacity. The theoretical analysis also demonstrated that, for policies for which approval was required at a Cabinet level (the *LiveLighter®* campaign, *Menu Kilojoule Labelling Legislation* and the *Land Use Planning Policies*), a collaborative, whole-of-government approach was required as part of policy decision-making [48,51]. However, in undertaking whole-of-government policy development processes, several competing forces emerged across the HTV policy process investigations. Firstly, whole-of-government processes resulted in increased *pressure from competing policy issues* (B3). For example, with respect to *Land Use Planning Policies*, whole-of-government consultation processes brought to light multiple other problem representations, such as the need for improved public transport infrastructure, actions on climate change, housing affordability, community and social infrastructure, and protection of agricultural land, among others. Interviewees indicated that this large 'problem load' had a substantial negative effect on the potential inclusion of mechanisms to support obesity prevention, with policy makers only able to attend to a limited number of priority issues. Secondly, whole-of-government consultation processes necessitated *involvement of a diverse range of stakeholders* (B4), including at both state and local government levels, which increased the complexity of policy processes. For example, according to participants, in order for the *Healthy Catering Polices* to be adopted, decision-maker support and formal approval was required at numerous hierarchical levels and across several departments of multiple organisations. This required policy brokers to convince executive managers of the benefits of the policy for each organisational unit.

Moreover, organisational rules meant that substantial consultation processes within each organisational unit were required before formally proposing the policy to decision-makers. The complexities associated with whole-of-government consultation often served to reduce support for proposed policy interventions from decision-makers. The nature of the required processes to adopt a whole-of-government approach to decision-making also substantially increased the *time required for the policy process* (B5). As explained above (B1 and B2), the long duration of policy processes reduced the likelihood of policy adoption. Moreover, the time taken for whole-of-government and external consultation reduced the opportunity for policy actors to learn from policy adoption success, thus creating several balancing loops that contributed to policy resistance.

## Discussion

This case study examined the dynamics underlying decisions regarding obesity prevention policy adoption in an Australian state government. The study articulated the complexity of obesity prevention policy decision-making processes, with multiple interconnected feedback

loops influencing the progress of policy proposals. The policy system was generally resistant to change, with five balancing loops acting to reinforce the policy status quo. These balancing loops including an organisational norm of risk aversion, and the complexities resulting from a whole-of-government policy approach and in-depth stakeholder consultation. However, seven virtuous reinforcing loops helped overcome policy resistance for several policy proposals. These virtuous reinforcing loops involved policy actor capabilities and capacity to develop viable evidence-based policy solutions, conduct effect policy development processes, and overcome political barriers such as stakeholder resistance. Policy actor capability to develop convincing policy proposals increased over time as policy actors gained experience in advocating for change and learned from other jurisdictions that had implemented obesity prevention policies.

The CLD generated as part of this study enhanced previously published theoretical analyses of obesity prevention policy decision-making systems by making explicit how underlying feedback loops either spurred policy change or resistance [61]. The CLD provided a way of representing the multiple, and often conflicting, dynamics of HTV policy decision-making [33]. The addition of a systems thinking approach to theoretical studies of obesity prevention policy decision-making provided additional insights regarding the potential leverage points, which may further assist the development of strategies and tactics to advance obesity prevention policy action in future. Key leverage points identified through the analysis are further discussed below.

## Capacity and capability of policy makers

The findings of this study highlighted the many policy system feedback loops that are influenced by the skills and capability of policy makers, both of which are central elements within the ACF and MST. When policy actors employed various strategic skills, a number of virtuous system effects were stimulated, such as the development of viable policy solutions that aligned with decision-maker beliefs and government objectives, as well as improved implementation feasibility and stakeholder support. Hence, in order to advance the implementation of obesity prevention policy solutions, it is of critical importance to ensure high political and policy capabilities of policy actors involved in obesity prevention policy.

Whilst the importance of policy actor capability is highlighted in many studies from other policy issues [56,62,63], this factor has not been frequently emphasised in previous obesity prevention policy studies [64]. A previous study that used CLDs to understand obesity prevention policy decision-making, by Waqa and colleagues [37], also found workforce capability to be important for obesity prevention policy adoption. Whilst these authors sought to improve this policy change determinant through formal training, advances in public health professional policy capabilities and political astuteness may also be achieved through reflective practice techniques, as well as organisational interventions that seek to reduce staff turnover [65,66]. Pooled resourcing among public health allies to increase public policy advocacy capacity and capability is also likely to be important in Australia and related contexts where there is typically a scarcity of funding and resourcing for the health promotion sector [67].

## Policy risks and organisational risk aversion

The findings of this case study demonstrated the central role of organisational risk aversion and the consideration of political risks within policy decision-making processes. Higher risk policies in the area of obesity prevention required substantial time to adopt, which created a threat to the likelihood of policy change in the context of short political cycles. For example, the long time period related to the extensive reviews of the *Menu Kilojoule Labelling Legislation*

meant that policy development processes occurred across a change of government, thereby creating complexity and delays in policy processes [48]. Consequently, this study highlighted the need to improve the skills of policy makers in effective negotiation and consensus building in order to facilitate timely progression of obesity prevention policy solutions. In addition, changes to institutional culture to one that supports innovation regarding preventive health is likely to be beneficial in securing adopting of recommended obesity prevention policies [68]. These findings are consistent with the public policy literature [68–70], and is an important insight within the obesity prevention context.

## Whole of government decision-making

Whilst there is a plethora of literature highlighting the need for whole-of-government policy development processes in order to improve the likelihood of obesity prevention policy adoption [71–73], this study demonstrated that the extended time required for a whole-of-government approach to policy development resulted in additional barriers to policy adoption. Several other studies have noted barriers to inter-sectoral decision-making related to preventive health, including siloed working practices, varied standards for evidence, and differences in organisational culture, priorities and incentives across different departments and sectors [74,75]. As noted recently [76], management of policy decision-making complexity requires more than single structural solutions, such as interdepartmental committees. Instead, a range of strategies are likely to be needed to reduce the complexity associated with the involvement of multiple sectors, particularly to address complex policy issues such as obesity, and to improve governance arrangements that can facilitate effective policy development.

## Stakeholder consultation

Like Waqa and colleagues [37], the current study found stakeholder consultation to be an important enabler for obesity prevention policy acceptance with decision-makers. However, unlike this previous study, we identified how the lengthy time involved in genuinely and effectively engaging with stakeholders posed a risk to policy adoption. Consequently, there is a need for strategies that can successfully elicit feedback, but in a way that does not unnecessarily delay policy processes and minimises the impact of stakeholder backlash on decision-makers. Increasing leadership capacity within government to better coordinate stakeholder involvement and set out clear policy parameters, terms of reference for consultation, and processes for responding to feedback may assist with these challenges [70,77].

Interestingly, opposition to policy adoption from the food industry was not a prominent feature of the policy processes investigated in this study. Indeed, the case study found the food industry to be a powerful influence in the *Menu Kilojoule Labelling Legislation* policy process only. This is at odds with a vast body of literature that has highlighted the food industry as a central barrier to obesity prevention policy progress [15,78,79]. The comparatively little evidence of influence from the food industry demonstrated in this case study is likely due to the nature of the policy instruments included in the study, many of which did not present a significant threat to the industry. However, it may also reflect that the study methods did not specifically set out to identify food industry influence on policy processes, which are known to take multiple and diverse forms [80].

This study nevertheless demonstrated how other private sector industries (e.g., construction industry with respect to *Land Use Planning Policies*) and stakeholder groups (e.g., mental health groups with respect to *LiveLighter®*) influenced the progress of several HTV obesity prevention policies. The combined case study findings indicate that public health advocates should consider the possibility that a broad range of policy actors, beyond the food industry,

may oppose obesity prevention policies. By better understanding stakeholder motives for their disagreement with proposals, policy entrepreneurs are likely to be better equipped to develop strategies that can reduce resistance to change [81].

## Policy-relevant evidence

This study, like many previous studies of obesity prevention policy [82–86], highlighted the role of evidence for securing decision-maker support for policy change. The meta-CLD illustrated the various forms of evidence that were important for facilitating obesity prevention policy adoption. These included evidence of the policy problem, policy instrument effectiveness, and implementation feasibility, each of which were important in shifting decision-makers' support for policy adoption. Nevertheless, the case study demonstrated that policies may be adopted in situations of minimal or uncertain evidence of effectiveness, such as in the implementation of *Menu Kilojoule Labelling Legislation* [48], although this process took several years. In relation to other policies, such as proposed restrictions on fast food outlets under *Land Use Planning Policies*, the lack of evidence of policy effectiveness was identified as a barrier to policy adoption. The ACF theory of policy processes helps to explain these findings, with this framework noting that 'policy-oriented learning' must occur in order for the beliefs of policy actors to change [87]. This can take considerable time as the ideological lenses of individuals influence their receptivity to evidence [88,89]. Consequently, efforts focusing solely on knowledge translation and/or building of scientific capabilities within government organisations are likely to fail to deliver policy change [90]. Instead, multiple tactics, coupled together, are more likely to be effective in seeking to implement obesity prevention policy. For example, policy entrepreneurs who reframe evidence based on the political context and the decision-makers they are seeking to influence may be more successful in securing policy change [55,90].

## System adaptability

The meta-CLD of policy decision-making systems developed as part of this study provided a snapshot of the dynamics over the duration of the HTV initiative and within the context of the geographical setting of Victoria, Australia. As CLDs represents complex adaptive systems, the elements documented in the CLD may change over time. Furthermore, the effect of the outlined feedback mechanisms can be reversed in response to external stimuli, to alter the dynamics from virtuous to vicious cycles of system behaviour. For example, the feedback effects of 'softening up' processes can be reversed if stakeholder support is low from the outset (e.g., if the obesity prevention policy is not evidence-based, public health stakeholders might voice disapproval), resulting in increased political risk which would then reduce decision-maker support [91]. Nevertheless, this study demonstrated the heuristic value of CLDs to help interrogate the root causes of policy system behaviours, with the model presented in this paper suitable for adaptation over time [92,93].

Whilst this study provided one representation of key feedback loops underlying obesity prevention policy adoption in the Victorian context, the theoretical grounding increases potential generalisability to other relevant settings. Moreover, the feedback loops are potentially generalisable to other policy issues, at least within Victoria, where the institutional and political factors are similar in nature to those investigated in this study [94]. Where context specificity prohibits generalisation [95], the outlined CLD approach may be used prospectively to support policy system understanding and advocacy strategy development through identification of policy system patterns and points of leverage [29].

## Policy implications and key contributions to the literature

This study builds upon previous obesity prevention policy literature by providing 'real-world' insight into the leverage points and trade-offs that need to be considered by those seeking to advance obesity prevention policy implementation. Several barriers to obesity prevention policy adoption have previously been identified, including powerful food industry lobbying, limited political skills and knowledge among the public health community, and institutional factors such as government 'silos' and political turnover [46,96]. To date, however, few studies have investigated the dynamics between the various influences on policy progress [37,97].

A key insight identified in the current study was the importance of obesity policy actor skills and capability, specifically the ability to develop and propose desirable policy solutions. While improved policy actor capability has also been recommended in previous obesity prevention policy studies [37,96,98], there remains a lack of empirical evidence regarding the skills on which to focus capacity building efforts. The current study goes some way to addressing this gap by having elucidated specific skills likely to be important, including the ability to frame policy solutions to decision-maker beliefs and broader government objectives, the skills to effectively engage stakeholders, and the utilisation of evidence to develop viable policy solutions. Nevertheless, further research is required to explore effective ways to enhance these skills within the obesity prevention context.

The study also highlighted the potential trade-offs [61] regarding various policy processes (e.g., whole-of-government processes and stakeholder engagement) that are typically cited as 'best practice' [72,99]. The study demonstrated that, while extensive stakeholder engagement helped to increase support for policy proposals, stakeholder engagement also introduced a risk to the timeliness and effectiveness of policy adoption by amplifying a number of divergent stakeholder views. Policy makers should be cognisant of this tension when designing stakeholder engagement processes, and consider appropriate structures and processes to effectively and efficiently engage stakeholders. For example, targeted or hybrid consultation approaches [100] could be considered, along with the establishment of clear parameters and processes for responding to feedback [70,77].

In addition, this study provided a nuanced view of the potential role of evidence as part of policy processes. Although policy makers, advocates and previous policy studies agree on the importance of evidence in obesity prevention policy, the focus of such discussion has largely been on evidence of the policy problem and effectiveness of policy solutions [18,97,101]. This study demonstrated the need for policy actors to employ evidence regarding multiple aspects of policy implementation in support of policy proposals. For example, evidence of feasible implementation of policy from other jurisdictions and related contexts was found to be valuable in increasing decision-maker acceptability and support.

## Strengths and limitations

The major strength of this study was the combined utilisation of political science theory and systems thinking tools to provide a more comprehensive understanding of both the influences on obesity prevention policy decisions and the interacting dynamics at play. The results were further strengthened through triangulation of findings across various qualitative data sources and methods (i.e., interviews, documents and observations) and through bringing together data from studies of various policy instruments within the HTV initiative. Additionally, the use of multiple theories of policy processes provided a broader perspective on policy-making, thereby expanding opportunities to understand the influences on policy adoption and resistance [102].

In regards to limitations of the study, data collection and initial analysis was conducted by one researcher (BC), which has the potential to introduce researcher subjectivity and bias. Several strategies were used to reduce potential bias, including the review of analysis outputs by a second researcher, using a pre-defined deductive codebook, triangulation of data across methods (e.g., interviews, documents and field observations) and data sources (e.g., multiple participants and policy processes), and utilisation of reflexive practices throughout the study [103]. The substantial number of interviews that were conducted, and the high participation rate from potential interviews further increased the reliability of the study. Nevertheless, the inaccessibility of some potential interviewees, including politicians and political advisors, may have limited the perspectives gained. Future studies that successfully engage a greater number of political actors would be particularly valuable. While the study also included a large number of documents as part of the analyses, with access to relevant documents facilitated by the placement of one of the researchers within DHHS, some relevant documents were not available to the researchers as they were classified as 'in-confidence' or were held by other government departments.

A further limitation of the study is that the meta-CLD was developed from retrospective data, collected for the purposes of understanding obesity prevention policy decision-making through political science theoretical lenses rather than explicitly through a systems perspective. Consequently, the systems model resulting from the analysis of data may be biased to reflect the elements outlined in the theoretical lenses and may fail to capture other potential system elements. However, as others have noted [24,25,104], there are strong commonalities between political science theories and systems thinking approaches that suggest the utilisation of this qualitative data for the purposes of CLD development is appropriate. The insights generated from the study may also be limited in that the meta-CLD included only those systems structures and interconnections that were consistently demonstrated across multiple HTV policy processes. Hence, central influences that occurred for one particular HTV policy process, but not in others, were not captured in the meta-CLD. However, findings specific to the processes for individual policy interventions are available elsewhere [48,49,51]. Future applications of the CLD methodology to various obesity prevention policy processes may help build evidence regarding the characteristics that are universal, or at least common, to various policy systems, as compared to those that are context or policy instrument specific (e.g., regulation compared to taxation). Furthermore, as a qualitative 'systems thinking' method, the meta-CLD does not purport to determine which factors are the strongest drivers of system behavior (i.e., which loops are most 'dominant' [105]). Other systems thinking methods, such as system dynamic modelling or social network analysis, could be used in future to assess which feedback loops most strongly impact policy system behavior [36,106].

Finally, as policy implementation was outside of the scope of the HTV policy process investigations, the CLD did not incorporate the effects of policy decision-making influences on policy outcomes. For example, the study did not consider whether the extensive stakeholder consultation had adverse effects on policy impact, through industry influence, which has been noted elsewhere [78]. Hence, the application of CLD methods to the study of policy implementation in future studies is encouraged.

## Conclusion

Policy processes for obesity prevention are complex in nature and resistant to change. This study used a combination of political science theory and CLD methods to develop insights into the barriers and enablers to obesity prevention policy change in a way that reflects the underlying dynamics of decision-making. The study identified a number of virtuous feedback

loops, including several capabilities of policy actors that can be recognised, nurtured and strengthened to improve the likelihood of beneficial policy change. These capabilities include policy skills, political astuteness, cross-sectorial negotiation skills, consensus building and stakeholder management. The identification of balancing feedback dynamics that contribute to policy resistance helped to highlight characteristics of organisational and policy systems that can be altered to better support obesity prevention policy change. These include strategies to facilitate effective and broad-based consultation, both across government sectors and external to government, implemented in ways that do not result in substantial delays in the policy process.

## Supporting information

**S1 Appendix. Number of interview participants and data collection timeframe for each Healthy Together Victoria policy study.**
(DOCX)

**S2 Appendix. Semi-structured interview guide.**
(PDF)

**S3 Appendix. Documents collected and analysed within each Healthy Together Victoria policy study.**
(DOCX)

## Acknowledgments

BC was hosted by the Victorian Department of Health and Human Services in conducting this research. The views and opinions expressed are those of the authors and not necessarily those of Deakin University or the Department of Health and Human Services.

## Author Contributions

**Conceptualization:** Brydie Clarke, Boyd Swinburn, Gary Sacks.

**Data curation:** Brydie Clarke.

**Formal analysis:** Brydie Clarke, Gary Sacks.

**Investigation:** Brydie Clarke, Gary Sacks.

**Methodology:** Brydie Clarke.

**Supervision:** Boyd Swinburn, Gary Sacks.

**Writing – original draft:** Brydie Clarke.

**Writing – review & editing:** Brydie Clarke, Janelle Kwon, Boyd Swinburn, Gary Sacks.

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
