## [Decision Letter · Decision Letter 0]

1 Dec 2020

PONE-D-20-21747

Understanding the dynamics of obesity prevention policy decision-making using a systems perspective: a case study of Healthy Together Victoria

PLOS ONE

Dear Dr. Sacks

Thank you for submitting your manuscript to PLOS ONE. I apologise for the delays in getting back to you. I have just recieved the second review required by the journal to enable an editorial decsion to be made.  After careful consideration, we feel that it has merit but does not fully meet PLOS ONE’s publication criteria as it currently stands. Therefore, we invite you to submit a revised version of the manuscript that addresses the points raised during the review process.

The first reviewer has indicated that minor revisions are required, in particular in relation to the reliability of the data analysis, which would benefit from  inter-coder relaibility assessment in relation to some of the data coded. Reviewer 2 has recommended rejection. However, if their concerns could be addressed, in particular in relation to greater discussion of  causes and current management challenges associated with the topic of obesity, togther with the limitations of cuurent managemnt practices, the MS would be of a standard accepatble for publication PLOS One. Please also ensure that the methodologies are presented in such a way that replication of your rersearch is feasible. 

We look forward to receiving your revised manuscript.

Kind regards,

Lynn Jayne Frewer, MSc PhD

Academic Editor

PLOS ONE

Journal Requirements:

2. In your Methods section, please provide additional information about the demographic details of your participants. Please ensure you have provided sufficient details to replicate the analyses such as:

a)  a description of any inclusion/exclusion criteria that were applied to participant inclusion in the analysis (specifying levels of experience) and

b) a table of relevant demographic details.

Reviewers' comments:

Reviewer's Responses to Questions

**Comments to the Author**

1. Is the manuscript technically sound, and do the data support the conclusions?

Reviewer #1: Yes

Reviewer #2: Partly

2. Has the statistical analysis been performed appropriately and rigorously? 

Reviewer #1: N/A

Reviewer #2: No

3. Have the authors made all data underlying the findings in their manuscript fully available?

Reviewer #1: Yes

Reviewer #2: No

4. Is the manuscript presented in an intelligible fashion and written in standard English?

Reviewer #1: Yes

Reviewer #2: Yes

5. Review Comments to the Author

Reviewer #1: This is a very well-written paper on a highly relevant and timely research topic. Many interventions and programs are available, but the key challenge is how to get them implemented. This paper examines the dynamics underlying policy adoption practices in Australia. I am not an expert in CLDs, but the authors convincingly show the added value of these diagrams in understanding the mechanisms of change in policy adoption processes. Very informative to read in this context of obesity prevention initiatives.

The study setting and data collection is clearly described and also the selection of the six interventions. One concern that I have is related to the data analysis. Data was collected by one of the researchers and it seems that this researcher also analysed all data and produced the diagrams. It is not clear how reliability of the analysis was assessed. It would have been good to have a second coder code/analyse the same data to prevent researcher bias. Figure 1 displays the way the data was analyzed, but it is unclear whether some findings are based on the individual analyses that are reported elsewhere or new analyses were done on the data.

Results of the study are extensively discussed. I miss a bit more concrete translation into recommendations for future policy implications. It is now described at a bit abstract level.

Reviewer #2: The topic of these research is obesity prevention policy adoption. The paper is very well and clearly written with the problem and issues clearly stated. The topic of obesity, its elements, causes and current management challenges are not addressed. The topic of obesity appears to be incidental to the subject at hand, which is an abstract exploration of the application of 3 political theoretical frameworks to documentation and qualitative stakeholder interview data relating to obesity prevention policy. Measured core elements of the analysis are not elucidated so are difficult to evaluate. The method itself is difficult to evaluate and is not replicable in its current form. The results do not report the qualitative stakeholder interview data, and instead provide a summary of findings, which is difficult to assess. The conclusions are not supported by sufficient data reporting. In more detail:

Introduction:

The challenge of obesity as a public health issue is well introduced, with the acknowledgement that progress in policy interventions to date is inconsistent, and that most interventions target the individual, with less emphasis on regulatory responses. The paper identifies and characterises system feedback processes using a case study design. Focus is largely on policy methods without explaining the complexity of the obesity issue per se. The paper would benefit from provision of more detail in this respect.

Method:

Focus is on the provision of adjunct activities to individuals. Interview sample size is robust, however the sample itself is not described in any detail. Snowball sampling is used, however this is not a rigorous approach to sampling, and while the various types of stakeholders are listed, no information is provided on the target samples for each type of stakeholder, if or how a balance of perspectives was sought, and how participants were selected beyond the fact that (1) snowball sampling was undertaken, and (2) selection was based on ability to provide detailed insight etc. How was ability assessed? How were stakeholder type numbers decided upon? Within what bounds? How long did the interviews last? Where did they take place? (How) were they recorded? How many researchers conducted the interviews? What steps were taken to standardise across researchers? How were the questions developed? There appears to be an inconsistent balance in the number of interviews around the 6 HTV initiatives, why was this? How will this affect the balance of perspectives? A wide and inconsistent range of documents are included, how will this affect the consistency of the findings?

Three political science theories are selected, however it is not clear how they are used in classifying data to develop key constructs. The reader is referred to another paper (Kim and Andersen) however it would be helpful if a brief explanation was to be given here.

Results:

These are reported logically and described clearly, however the findings appeared very abstract with no specific supporting evidence provided (e.g. information source in each case: interview or documentation). Information is reported for what was found, however it is not clear how it was found. While the narrative is interesting, it is not clear how the findings were arrived at.

Given the central importance of capacity and capability of policy makers to the arguments summarised in the discussion, how was capacity and capability of policy makers characterised by participants? Results refer to ‘Government policy makers or external policy actors with sufficient capability were able to frame policy narratives to align policy proposals to decision-maker beliefs’ – how was this evaluated? It is not clear how these inferences were arrived at given the scope and content shown in the Appendices.

Discussion and Conclusion:

Extensive and interesting, however I do not understand how the conclusions were reached given the sparse nature of information provided concerning the methodology. While the case study used related to obesity prevention, there is little in this section which relates to the challenges of obesity prevention, or how obesity prevention differs from other public health issues. This section therefore comes across as very generic, e.g. referring to public health policy in general as opposed to obesity prevention in particular. Previous studies are mentioned, but little information is provided as to how the current findings relate to these earlier studies. Conclusions are difficult to assess given the lack of detail provided relating to the underlying data, and there is insufficient evidence to suggest that the research has been conducted rigorously.

6. PLOS authors have the option to publish the peer review history of their article (what does this mean?). If published, this will include your full peer review and any attached files.

Reviewer #1: No

Reviewer #2: No

---

## [Author Response · Author response to Decision Letter 0]

18 Dec 2020

Please refer to response to reviewers document attached.

---

## [Editor Report · Decision Letter 1]

4 Jan 2021

Understanding the dynamics of obesity prevention policy decision-making using a systems perspective: a case study of Healthy Together Victoria

PONE-D-20-21747R1

Dear Dr. Sacks 

We’re pleased to inform you that your manuscript has been judged scientifically suitable for publication and will be formally accepted for publication once it meets all outstanding technical requirements.

Kind regards,

Lynn Jayne Frewer, MSc PhD

Academic Editor

PLOS ONE

---

## [Editor Report · Acceptance letter]

11 Jan 2021

PONE-D-20-21747R1 

Understanding the dynamics of obesity prevention policy decision-making using a systems perspective: a case study of Healthy Together Victoria 

Dear Dr. Sacks:

I'm pleased to inform you that your manuscript has been deemed suitable for publication in PLOS ONE. Congratulations! Your manuscript is now with our production department. 

Kind regards, 

on behalf of

Dr. Lynn Jayne Frewer 

Academic Editor

PLOS ONE